# Contextual, Client-Centred Coaching Following a Workshop: Assistants Capacity Building in Special Education

**DOI:** 10.3390/ijerph18126332

**Published:** 2021-06-11

**Authors:** María José López-de-la-Fuente, Pablo Herrero, Rafael García-Foncillas, Eva Mª Gómez-Trullén

**Affiliations:** 1Department of Physiatry and Nursing, Faculty of Health Sciences, University of Zaragoza, 50009 Zaragoza, Spain; mjolopez@unizar.es (M.J.L.-d.-l.-F.); evagomez@unizar.es (E.M.G.-T.); 2Department of Microbiology, Preventive Medicine and Public Health, University of Zaragoza, 50009 Zaragoza, Spain; rafagfl@gmx.com

**Keywords:** coaching, fidelity coaching practices, paraprofessionals/special needs assistants, school-based occupational therapy, participation

## Abstract

Assistants serve an essential role in special education to support children with disabilities, but they should be properly trained and supervised. The coaching approach represents one trend that has been gradually implemented in occupational therapy (OT) and rehabilitation services. Still, few studies clearly define the coaching intervention, measure the fidelity of coaching practices, or evaluate capacity building of the caregivers in the long term. This quasi-experimental study compared one-on-one coaching in natural environments following a workshop with a training workshop. Both public schools do not have regular OT services. The primary outcome was the assistant’s performance, measured with the Goal Attainment Scaling (GAS). The secondary outcome was the fidelity of coaching implementation, measured with the Coaching Practices Rating Scale (CPRS). The GAS showed an increased performance of the assistants after the intervention, with significant differences between groups post-intervention (*p* = 0.015) and large effect size (*r* = 0.55), but no long-term significant improvements were found at the follow-up (*p* = 0.072). The CPRS showed an adequate implementation of the five coaching components (joint planning, observation, action, reflection, and feedback), with a total score of 3.5 ± 0.72 (mean ± SD). The results suggest that coaching sessions provided by OTs in schools may improve assistants’ skills to facilitate the student’s participation.

## 1. Introduction

Functional independence in self-care activities is crucial to ensure community participation of people with severe/multiple disabilities [1]. This participation is not exclusively influenced by the severity of motor or intellectual disability, but it can be affected by environmental factors as well [2]. Therefore, focusing on occupations and eliminating environmental barriers or creating facilitators may have a greater impact on the promotion of the students’ participation, since some personal characteristics may not be modifiable, unlike other environmental factors [3,4,5]. For this reason, recent studies have highlighted that these occupational therapy (OT) interventions should consider all the person–occupation–environment aspects together [3].

Although it is known that it is possible to work on the aforementioned environmental factors in school settings, school-based OT still remains an emerging practice in Europe [6,7]. Although OT’s best practices in school demand the adoption of an ecological approach [8], school-based OTs seem to prefer direct intervention in separate settings, in stark contrast to current research, which recommends working through collaborative, consultative, and contextual models [5,9,10,11].

Apart from OTs, there are many other professionals and paraprofessionals/assistants who work in inclusive and special education. Focusing on the assistants, researchers question to what extent this staff can improve the students’ development [12,13,14,15] or facilitate their participation [13,16]. Although assistants currently play an essential role in supporting personal care activities, some studies also point out that children who have an assistant seem to perform poorer in school activities. However, the reason for this lower performance is not clear [17].

Regarding the controversial contribution of assistants in school settings, recent studies highlight that future challenges should clarify their functions, develop plans to improve their work [14,18,19], and provide opportunities for them to collaborate with team members [8,9,12,13]. For this reason, optimal training and follow-up procedures should be developed and the generalization and maintenance of skills should be assessed [15,20,21].

Without a doubt, adequate training of the school staff is essential to support students with disabilities. One of the most commonly used methods to teach school staff is instruction through training workshops, although other complementary methodologies such as coaching have shown to be useful for assistants [15,20]. The coaching approach can encourage clients to develop strategies to overcome environmental barriers autonomously [22] and therefore help the assistants to support students’ participation.

Although several studies recommend coaching as an intervention strategy in OT, many of them do not make a clear definition of this approach or analyse the involved components [23,24,25,26,27]. For these reasons, the OTs that use coaching practices need to involve critical components, which have been already pointed out in the recent literature: (1) client-centred, (2) collaborative, (3) reflective, (4) promoting capacity, (5) ecological (taking place in the client’s natural environment), (6) strength-based, (7) promoting self-determination, (8) using positive language, and (9) focusing on a preferred future or goal [23].

To the best of our knowledge, few studies have compared different training types and their outcomes among assistants. Moreover, most of these studies do not have a control group and are limited to analysing the short-term effects. Therefore, the main objective of this study was to compare whether coaching sessions following a workshop are more effective than just a training workshop in order to improve the capacity building in the assistants in the short and long term. Furthermore, current research indicates that therapists should describe and document their adherence to coaching key ingredients [23,25], and because of this, our secondary objective was to examine the implementation fidelity of coaching practices.

## 2. Materials and Methods

### 2.1. Study Design

This quasi-experimental study, performed with a non-probabilistic convenience sampling, was conducted in two of the five public special education schools in Zaragoza (Spain) during the 2016–2017 school year. We used a pre-and post-test non-equivalent control group design. The Clinical Research Ethics Committee of Aragón (CEICA) approved this study protocol (CP-CI PI16/0247). The study design followed the standards of the Declaration of Helsinki for biomedical research. The study was registered at clinicaltrials.gov with number NCT04747210.

### 2.2. Participants and Setting

All available assistants from both schools were invited to participate. The participants from the first school were allocated to the intervention group, whereas the participants from the other school were allocated to the control group in order to avoid potential bias when performing the intervention. The first school had 20 special need assistants and 99 students, while the second one had 11 assistants and 64 students. Like the rest of the public schools in the city, none of them had a regular OT service, and, therefore, this was a new service included in this study. The inclusion criteria for assistants were as follows: (1) being available for the workshop and one-on-one coaching and (2) written consent. The inclusion criteria for students were as follows: (1) presenting difficulties in activities of daily living, (2) having an assistant involved in the study, and (3) informed consent signed by parents or legal tutors.

### 2.3. Measures

Sociodemographic data were gathered with ad hoc structured questionnaires/surveys, including the Care Dependency Scale for Pediatrics (CDS-P) in the case of students [28,29] in order to determine their degree of dependence. The CDS-P has 15 items scored on a five-point Likert scale, ranging from 1 (completely care-dependent) to 5 (almost independent). The overall score ranges from 15 to 75 points, and lower scores indicate more dependence than higher scores. The performance was measured with the Goal Attainment Scaling in the baseline (GAS-1) and the end of the school year (GAS-2), whilst the fidelity of coaching was measured with the Coaching Practices Rating Scale after the intervention through recorded videotapes.

The primary outcome measurement was the Goal Attainment Scaling (GAS) [30], which is an individualized and client-centred measure that enables the monitoring/comparison of performance of a client or groups over time [31]. GAS is an inherently flexible assessment that can be applied with minimal cost and allows involving clients in the learning process. It is recommended for the measurement of coaching outcomes in research and practice [32]. Firstly, clients identify a set of goals. The weight of each goal is calculated by multiplying its importance (ranging from 1 = a little important to 3 = really important) by its difficulty (ranging from 1 = a little difficult to 3 = very difficult). Afterwards, clients rated their performance in each goal on a 5-point Likert scale ranging from −2 (much less than expected) to +2 (much better than expected). In the baseline, clients could score −1 or −2, and the score “0” is the desired level of success. Goal-attainment levels and the relative weights allow the computation of the T-score with a mean of 50 and a standard deviation of 10 [30].

The Coaching Practices Rating Scale [33,34] was utilized as a secondary outcome to measure the fidelity of coaching. Its 14 items are based on the adults’ learning characteristics and coaching practices with the aim of determining the extent to which the coach uses these practices. This scale accepts the rating through video tapes and has been shown to have a high degree of construct validity and internal consistency [35]. For each item, the evaluator indicates how often the practice was used by the coach on a 6-point Likert scale: 0 = No opportunity to measure or use the practice, 1 = none of the time, 2 = some of the time, 3 = about half the time, 4 = most of the time, 5 = all of the time. A special characteristic of this scale is that a rating of 1 indicates that the described behaviour was not used by the professional even though the opportunity occurred. The total score can be used to measure adherence to coaching practices [35].

Moreover, 18 months after the end of the intervention, all assistants were invited to reply to a semi-structured interview through a phone call. This interview collected information about their current employment, the degree in which they had been able to put into practice the skills learned, and the challenges for their professional development. Two open-ended questions were asked to get a deeper perspective of the assistants: “Please give an example if you have been able to put what you have learned into practice” and “Please explain the main challenges in your job”.

### 2.4. Procedures

#### 2.4.1. Intervention

Both schools collaborated to implement training and follow-up procedures. A ten-hour training workshop titled “Improving participation in school: creating learning opportunities” was performed outside working hours, divided into three sessions. Afterwards, the OT accompanied the assistants of both the control and the intervention group during a workday morning. These meetings allowed us to analyse the difficulties in the participation in daily occupations, to record routines such as schedules and places, and to ask reflexive questions about the challenges observed. Assistants identified three GAS goals related to and aligned with the students’ goals (e.g., decrease verbal or physical aids in dressing/toileting/eating; enable children participation and diminish load in functional mobility/transfers; give children time to respond to activity demands; encourage learning new tasks or refine existing skills; facilitate performance in daily occupations selected).

Additionally, unlike the control group, the intervention group received one-on-one coaching, conducted by an experienced OT, from January to March (10 weeks). Coaching sessions were recorded and adapted to every assistant’s needs and focused on increasing child–caregiver interactions and child-learning opportunities in everyday routines and contexts. The five coaching practice characteristics of (1) joint planning, (2) observation, (3) action/practice, (4) reflection, and (5) feedback, identified by Rush and Shelden, were utilized [34,36]. In this interactive process performed in this context, the coach guided the caregiver, encouraging self-discovery and personal development [36]. At the end of the coaching program, the assistants had between two and three months to apply the strategies learned.

#### 2.4.2. Follow-Up

At the end of the academic year, the assistants assessed their current level of performance with the GAS. In order to measure the fidelity of coaching practices, an external OT evaluated 25% of the videotapes, which were randomly selected previously. After 18 months, a follow-up consisting of phone calls was carried out. An external evaluator analysed the answers, which had been recorded. Some responses were directly transcribed due to their descriptive nature.

### 2.5. Data Analysis

Analyses were conducted using Statistical Package for the Social Sciences (SPSS) version 26.0 and Microsoft Excel 2010. To examine the demographic characteristics of the students and their caregivers in both groups, we conducted independent t-tests and chi-square tests for continuous and categorical data, respectively. The Kolmogorov–Smirnov and Shapiro–Wilk tests were used to determine whether the data were normally distributed. Data that were not normally distributed were analysed using the Mann–Whitney U test or Wilcoxon signed-rank test. To investigate the relationship effects of the intervention, we compared the between-group variations for the performance of the GAS-1 (baseline) and the GAS-2 (post-intervention) using T-scores. The percentages of the responses to each question of the assistants’ questionnaire/call were analysed. The mean of the 14 items in the Coaching Practices Rating Scale and the five coaching characteristics were examined. The criterion for significance of the findings was set at *p* < 0.05. The effect sizes were evaluated using the guidelines proposed by Cohen (1988), considering the following intervals for *r*: 0.1 to 0.3 (small effect); 0.3 to 0.5 (intermediate effect); and 0.5 and higher (strong effect) [37].

## 3. Results

Nineteen assistants were recruited. Two of them did not complete the intervention procedures (one assistant of the intervention group left the employment; one assistant of the control group did not have any students enrolled). Therefore, there were 13 assistants in the intervention group and 4 assistants in the control group. The student–assistant ratio is about the same in both schools (4–6 students per assistant); only one assistant in the intervention group works part-time and supports a student. There were no significant differences between groups in terms of age, gender, marital status, or educational level. However, there were differences regarding the time they had been working as assistants (*p* = 0.005); in the control group, they had more years of experience. Only 17.6% of the assistants had received specific training before they started working. All the assistants considered that their functions were not well delimited. Furthermore, only 29.4% of them had been trained in personal care (Table 1).

Forty-two families were assessed for eligibility; one student of the intervention group was excluded because he did not meet the inclusion criteria. Moreover, four students dropped out of the program due to health problems, two of them from the intervention group.

Therefore, the sample consisted of 24 and 13 students in the intervention and the control group, respectively. There were no significant differences regarding the age, gender, or medical diagnosis of the students between the groups. However, significant differences were found concerning the educational diagnosis (*p* = 0.005). In the case of the control group, 23.1% of the students were diagnosed with multi-deficiency, whereas the percentage in the intervention group was 70.8%. No significant differences were found in terms of levels of dependence in the CDS-P (*p* = 0.054) (Table 2).

Mean T-scores of each group in the GAS were used to evaluate the training outcomes. At baseline (GAS-1), no statistically significant differences were found regarding the performance of the assistants (*p* = 0.350). At post-intervention (GAS-2), the intervention group had higher scores than the control group, with significant differences concerning the performance (*p* = 0.015), and the observed effect size was large (*r* = 0.55). Figure 1 provides a graphic comparison of the GAS T-scores before and after the intervention.

The number of sessions delivered was between 3 and 8 (mean = 5.8). All the coaching sessions were recorded. However, neither the daily routines nor the meetings with the groups of assistants were recorded. The average time of the coaching sessions was 15 min, although the duration depended on the availability of the assistants and the occupations. It is important to remark that many conversations with the assistants took place outside the scheduled sessions (e.g., in the hall or during the coffee break).

Each one of the two assistants who had received 7 or 8 sessions supported only one participant student enrolled, and these two students presented a severe dependence measured with the CDS-P (19 and 18 points, respectively). Another assistant, who had received six sessions, had three students enrolled, and they also presented severe levels of dependence (15, 15, and 19 points). These three assistants had more than six years of experience, and at least one of their GAS goals was to improve the child’s participation in transfer from the floor to the chair and vice versa.

Sixty-seven videotapes were collected. Six of them had to be discarded due to technical problems (e.g., recording stopped in the middle of the session). The sample consisted of 61 videos, from which 25% (15/61) were analysed. The five coaching characteristics were implemented by the OT during the sessions (mean total score = 3.5), although the implementation was not carried out to the same extent (Table 3). The lowest score was obtained in questions 3, plan coaching (mean = 2.5; SD = 1.25); 11, comparative questions (mean = 2.9; SD = 1.10); and 14, evaluate coaching (mean = 2.9; SD = 1.33). However, consideration of adult learning (items 1–2) received the highest scores (mean = 4.1). Observation and practice reached a mean of 3.9 points; feedback had mean = 3.4, reflection had a mean = 3, and joint planning had the lowest scores with a mean of 2.9.

Eighteen months after the intervention, over 88% of the participants kept on working as Special Needs Assistants. Nevertheless, 61.5% of intervention group assistants no longer worked at the same school, so significant differences were found between groups (*p* = 0.031). All the control group assistants had continued working in the same position. However, in the intervention group, one assistant worked in a geriatric home, and another one had been unemployed during this time frame. Furthermore, six caregivers had moved to other schools, and four of them were transferred to inclusive classrooms. A total of 64.7% of the assistants attended various courses after the program’s end; these courses were theoretical and took place outside working hours. Moreover, 82.4% recognized that practice, experience, and personalized training would be the best option for their professional development.

No statistically significant differences were found regarding the transfer and generalization of learning (*p* = 0.072), although a medium effect size was obtained (*r* = 0.43) (Table 4). However, we consider that it is relevant to reflect some extracts from the assistants’ calls: some special education schools’ assistants explained that the coaching program had been helpful to facilitate the participation of the students in different occupations. Now, they give the children more time and allow them to demonstrate what they can do. Furthermore, they seem to have new ideas and feel more self-reliant. However, most of the assistants who work in inclusive schools commented that they were supporting very different students with other problems and argued that they would need another kind of training to help these students. Finally, many of them also indicated that they would like the OT to go back to their school.

These comments reflect primarily the need to train and supervise assistants according to their roles. Moreover, on the other hand, although they were willing to support students with disabilities, they must learn how to facilitate the participation of these students.

Concerning the challenges they face in their daily work, the assistants emphasized the need for adequate training (personalized and practical), focused on their everyday problems and guided by other professionals, and they valued the coaching program positively. They commented that the lack of delimitation of their functions and other organizational aspects (such as substitutions, differences in working hours, and student–assistant ratios variability), hinder their work. Finally, they believe that their work is not sufficiently valued. These comments suggest that day-to-day problems for assistants continue being there and should be taken into account.

## 4. Discussion

Our study compared which kind of training for assistants can obtain better results. The findings support that the context approach and one-on-one coaching following a workshop are more effective than only a workshop in order to improve the performance of the assistants. Therefore, the school-based OTs should strive to create contextual interventions that are occupation-centred and goal-oriented.

As in previous research, the participants in our study believe that their functions are still not well delimited and that they require training adapted to their real needs, as well as that their work is not sufficiently valued [13,14,15]. Moreover, and similarly to previous studies, our data show that the assistants have a scarce training level and that the training courses are excessively theoretically oriented, preferring continuous and practice-based training to update their knowledge [19,21,38]. Our study confirms that there is still a gap, already seen more than a decade ago, between the assistants’ training and their roles within the school.

Several authors point out that the main barriers for an effective collaboration are the lack of time, the receptiveness to work in team, and the assumption of the expert role by the therapist [11,39]. In our study, these challenges in collaboration were minimized through the following factors: (a) the OT created deliberately collaborative relationships with the assistants; (b) the OT was available to address any concerns of the assistants, as in Hutton’s pilot study [40]; (c) the OT and the assistant worked together to find effective solutions. We consider that these factors may have had a positive influence on the outcomes. Moreover, the attention to diversity in school requires individualization and ongoing teamwork [13], which was achieved in our study through the identification and individualization of objectives performed by both assistants and OT. This resulted in the implementation of strategies that we could not have imagined to support children’s self-care in daily routines, such as the rapid achievement of the goal, which was also seen in other studies with parents [4,41]. Like other researchers, we consider that valuing the assistants’ work, training and supervising them properly, and giving them a voice within the team is essential to minimize negative attitudes and overdependence [13,42].

We would like to point out that, in our study, the assistants showed a high engagement in the coaching process. This fact contrasts with previous research, where the assistants refused to participate in the coaching sessions (Walker et al., 2017). This high engagement could be due to the fact that our intervention was performed just-in-time (in real context) and that the participation was active, bringing about better learning outcomes [43].

The assistants achieved noticeable improvements that allowed children with multiple disabilities to have more time and opportunities to participate in everyday occupations, such as the development of skills to facilitate swallowing, feeding, grooming, dressing, or functional mobility, and they understood that some environmental barriers could be easily removed. Our results were similar to other studies conducted with parents or teachers, where coaching interventions were effective to increase capacity building [41,44,45]. Moreover, the assistants discovered new ways of doing things and came up with innovative ideas such as using a toothpaste dispenser after seeing the automatic soap dispenser or replacing shoelaces by other simple aids. Furthermore, they reflected on their actions (e.g., “This week I discovered that this way we do it better”; “I have to organize myself better”) and they realized that students with severe disabilities can participate beyond than expected as the program began (e.g., “this boy has surprised me; now he helps me and many times he places by himself”, “this girl can do more things than I expected”). Therefore, educating assistants can diminish attitudinal barriers and change negative perceptions to facilitate children’s participation [5].

Regarding the number of sessions to complete the process, there were differences between the assistants, since some of them needed more sessions than others. The years of experience did not seem to have influenced the number of sessions required. However, our data indicate that the students’ level of dependence, the personal characteristics of assistants and the goals’ nature must be considered when scheduling the coaching sessions. Therefore, the coaching procedure is not necessarily linear and must be adapted to the coachees’ changes and real needs [36]. Therefore, it is also necessary to have some resilience when using the coaching approach in research or clinical practice, as it may be challenging for the OTs and researchers. This is in line with authors such as Rush and Shelden, who defend the claim that therapists should be flexible and empower stakeholders to identify problems, goals, and solutions when they apply the coaching approach in natural environments [34].

On the other hand, our findings suggest that it is necessary to assess the changes in participants’ capacities and to measure the fidelity of the coaching practices. We used reliable tools such as the GAS [32] and the Coaching Practices Rating Scale [33,34], as suggested by other researchers who defend documenting changes in the coaching recipients’ competence objectively and reporting the fidelity of the implementation of coaching practices [25,27]. Thanks to the objective measurement of the intervention, we realized that the OT used positive language and the intervention focused on participants’ preferred goals through the GAS. The results of our study differ from previous reviews, which highlights that both the use of a positive language (6% of the studies) and preferred goals (12% of the studies) were the least used [23]. Thus, OTs need to analyse how they use coaching elements, as recommended by several authors [23,25,34], to remove the discrepancy between theory and practice and apply the coaching approach in OT more reliably.

Long-term follow-up of participants (parents or caregivers) has been recommended in previous research [25]. In our study, despite carrying out a long-term follow-up, we did not find statistically significant differences in terms of generalization and transfer of learning, although the responses from several assistants were encouraging. The explanation may be that some assistants from the intervention group had changed their job and moved to inclusive schools. These assistants remarked that some of the learned competences could not be applied to the students who did not have any performance problems in self-care. This fact cannot be considered trivial; the assistants’ roles can vary even within the same school [15]. In sum, assistants’ and students’ needs may be different over time and in different contexts. Therefore, the individualized strategies for each child and caregiver [4], the ongoing supervision and training of assistants [12,13,19], and the implementation of OT’s services in the school [7,8] must be taken into account to achieve meaningful outcomes.

### Strengths and Limitations

Our study has some strengths, such as measuring the fidelity of the coaching practices and the fact that participants have been allowed to select their preferred goals. Moreover, we carried out a follow-up 18 months later, which is a gap detected in previous research. However, we also had some limitations, such as not randomizing participants, although this was done to avoid cross-contamination among participants. Furthermore, our sample was small, although the greatest limitation of this study was convenience sampling and unequal groups. Other contextual factors may have influenced the final results in different ways, which should be considered in future studies, such as the internal organization of each school, as well as the number of employees and their characteristics (e.g., years of work experience), among others. Future studies should be carried out with more participants and an RCT design to allow the generalization of the results. Finally, we believe that more research is required to generate robust evidence to support the future implementation of OT services in Spanish schools.

## 5. Conclusions

Our study shows that a coaching approach provided by OTs in schools may improve assistants’ skills to facilitate children’s participation in daily occupations. Coaching sessions following a workshop can help assistants put their abilities into practice to support students with disabilities. Personalized interventions such as one-on-one coaching are necessary due to the wide variety in the students and assistants.

School-based OTs should take into account the ecological factors, the assistants’ and students’ challenges, and the adult learning characteristics to design appropriate training packages. Coaching performed in the context may be effective to ensure that children with disabilities and their caregivers participate in rehabilitation programs in te school settings.

## Figures and Tables

**Figure 1 ijerph-18-06332-f001:**
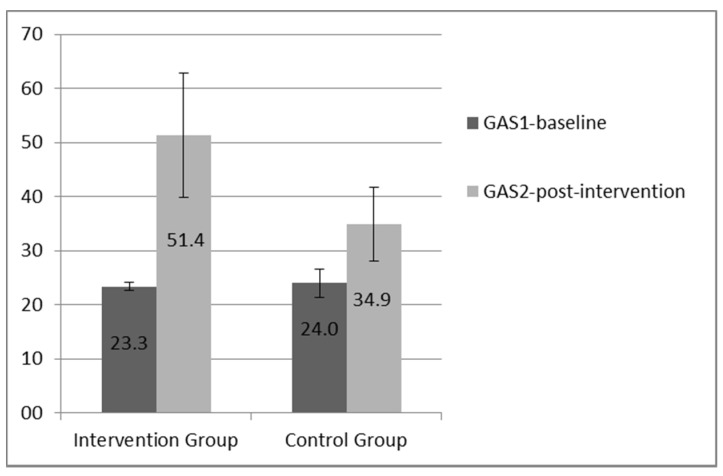
Mean GAS T-scores pre- and post-intervention. Intervention G: GAS1 mean = 23.3 (SD 0.74); GAS2 mean = 51.4 (SD 11.44). Control G: GAS1 mean = 24.0 (SD 2.58); GAS2 mean = 34.9 (SD 6.83).

**Table 1 ijerph-18-06332-t001:** Demographic Characteristics of Assistants.

		n = 17	IG (n = 13)	CG (n = 4)	*p*
Chronological age ^a^		47.35	45.54 ± 7.54	53.25 ± 8.26	0.1 ^b^
Gender %	Female	94.1	100	75	0.235 ^c^
	Male	5.9	0	25	
Marital status%	Single	17.6	15.4	25	0.65 ^c^
	Married	82.4	84.6	75	
Educational level %					
Primary Education		5.9	0	25	0.199 ^c^
Vocational Education		23.5	30.8	0	
Baccalaureate		17.6	15.4	25	
University Degree		52.9	53.9	50	
Years of work experience%	<1	17.6	23.1	0	0.005 ^b^
2–6		11.8	15.4	0	
7–10		23.5	30.8	0	
11–19		29.4	30.8	25	
>20		17.6	0	75	
Previous training%	Yes	17.6	15.4	25	0.579 ^c^
No		82.4	84.6	75	
Personal Care Training%	Yes	29.4	30.8	25	0.670 ^c^
No		70.6	69.2	75	
Your functions at school are well defined?%					
No		100	100	100	-

^a^ mean ±standard deviation. ^b^ Independent *t*-test; ^c^ Chi-square test. Note: IG = intervention Group. CG = Control Group.

**Table 2 ijerph-18-06332-t002:** Demographic Characteristics of Students.

		n = 37	IG (n = 24)	CG (n = 13)	*p*
Chronological age ^a^		11.7 ± 5.1 4–21	12.66 ± 5.1 4–21	10.07 ± 4.9 5–17	0.14 ^b^
Gender %	Girls	24.3	29.2	15.4	0.44 ^c^
	Boys	75.7	70.8	84.6	
Medical Diagnosis %					
Cerebral Palsy		43.2	58.3	15.4	0.070 ^c^
Autism Spectrum Disorder		13.5	12.5	15.4	
Developmental Delay		21.6	16.7	30.8	
Down Syndrome		8.1	0	23.1	
Angelman Syndrome		5.4	4.2	7.7	
Intellectual Disability		2.7	0	7.7	
Lennox–Gastaut Syndrome		2.7	4.2	0	
Wolf–Hirschhorn Syndrome		2.7	4.2	0	
Educative Diagnosis %					
Multiple Deficiency		54.1	70.8	23.1	0.005 ^c^
Generalized Behavior Disorder		13.5	12.5	15.4	
Developmental Delay		18.9	16.7	23.1	
Intellectual		13.5	0	38.5	
CDS-P ^a^		33.5	29.96 ± 13.7 15–59	40.31 ± 17.4 16–67	0.054 ^b^

^a^ mean ± standard deviation. Minimum-Maximum ^b^ Independent *t*-test; ^c^ Chi-square test. Note: IG = intervention Group. CG= Control Group.

**Table 3 ijerph-18-06332-t003:** Descriptive Statistics Fidelity Coaching Practices.

	Scale Item	Mean	SD		Mean	SD
1	Acknowledge ability	3.6	1.35	Adult learning (1 + 2)	4.1	0.7
2	Non-judgmental interactions	4.5	0.74			
3	Plan Coaching	2.5	1.25	Joint planning (3 + 4)	2.9	0.6
4	Plan action	3.3	1.05			
5	Observe knowledge	3.9	0.88	Observation (5 + 6 + 7)	3.9	0.5
6	Observe skills	4.4	1.06			
7	Observe Coach	3.5	0.92			
8	Multiple practices	4.1	1.13	Action/Practice (8 + 9)	3.9	0.4
9	Learner practice	3.6	1.06			
10	Probe questions	3.1	0.83	Reflection (10 + 11 + 14)	3.0	0.1
11	Comparative questions	2.9	1.10			
12	Provide Feedback	3.1	1.39	Feedback (12 + 13)	3.4	0.5
13	Provide information	3.7	0.96			
14	Evaluate Coaching	2.9	1.33			
	TOTAL SCORE	3.5	0.72			

**Table 4 ijerph-18-06332-t004:** Results of Control Call Special Need Assistants (SNA).

		n = 17	IG (n = 13)	CG(n = 4)	*p*
Do you work as an SNA?	Yes	88.2	84.6	100	0.404 ^a^
	No	11.8	15.4	0	
Do you work at the same school? %	Yes	52.9	38.5	100	0.031 ^a^
	No	47.1	61.5	0	
Work setting %	Special education school	64.7	53.8	100	0.415 ^a^
Inclusive school		23.5	30.8	0	
Other (Geriatric home)		5.9	7.7	0	
Unemployed		5.9	7.7	0	
Training in past 18 month	Yes	64.7	53.8	100	0.091 ^a^
	No	35.3	46.2	0	
Type of training	Theoretical	64.7	53.8	100	0.056 ^a^
	Not applicable	35.3	46.2	0	
Training outside working hours.	Yes	64.7	53.8	100	0.056 ^a^
	Not applicable	35.3	46.2	0	
How do you think you learn more in your work? %					
	Practical and experience	5.9	0	25	0.096 ^a^
Personal training focused on my real needs		11.8	7.7	25	
	Both	82.4	92.3	50	
Indicate the degree to which you have put the skills learned into practice in your current position.					
	Mean	3.12	3.46	2	
	1 = nothing %	11.8	7.7	25	0.072 ^b^
	2 = very little %	29.4	23.1	50	
	3 = something %	17.6	15.4	25	
	4 = quite %	17.6	23.1	0	
	5 = a lot %	23.5	30.8	0	

^a^ Chi-square test. ^b^ Mann–Whitney U test.

## Data Availability

The data presented in this study are available on request from the first author.

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
