# Peer review of "Contextual, Client-Centred Coaching Following a Workshop: Assistants Capacity Building in Special Education"

_ijerph, 2021, doi:10.3390/ijerph18126332_

Round 1

Reviewer 1 Report

The purpose of this quasi-experimental study is about client-based training provided by occupational therapist for assistant in special education.The intervention approach included context approach and one-on-one coaching. The authors report that coaching sessions adapted with assistance's need  following a workshop are more effective than a training workshop in improving the working capacity of special Need Assistants.

This reviewer appreciate the work by the authors in this clinically important population to improve healthcare service among children with disability. However, there are several concerns while interpreting the study results.

Abstract

  1. The abstract could be more specific and concise to include outcome measurements and main results with data.
  2. The author mentioned about “needs of Spanish assistants are similar to those of other assistants” is not coherent with the main paragraphs.
  3. Apart from the training programs included in study intervention, how did the assistant collaborate with occupational therapists? Was there any difference between intervention and usual practice?

Method

  1. Although the authors mentioned about the group allocation was done to avoid cross-contamination. Potential bias may exist since the huge discrepancy of the assistant number from each group, variability of two special education schools, and years of working experience of the assistant from two group.
  2. The ratios of children to assistant are different between schools. Furthermore, the loading from taking care of children of each assistant was different. Was it possible that the loading influenced on the assistance’s performance?
  3. Despite on-
  4. No mention of the length of the training intervention and how long did the assistant receive one-on one coaching?
  5. How did the sample size determine? Is there any preliminary data or previous literature to support? And how was the effect size of GAS performance calculated?

Result

  1. Suggest predigesting the extracts and responses from assistant about generalization of learning and challenge from work.

Figure

  1. Figure 1. suggest revising to report standard deviation.

Reviewer 2 Report

The study is interesting in that the results show that coaching sessions following a workshop can help assistants put into practice their abilities to support students with disabilities especially in the school settings.

However, the manuscript should consider the following concerns to make the manuscript more effective and meaningful to the field.

Since the lack of the participants and the differences of the participants (i.e., 13 assistants in the intervention group and 4 assistants in the control group) in the study, I suggest that the authors should try the multiple imputation method in order to maximize use of available information (or data).

Even though the authors used the Mann-Whitney U test since the numbers of the participants of the intervention and control groups are different and the difference in numbers of the two groups is a rather big. However, I guess it would be better to use the test for homogeneity or normality test as well even though the sample was small with unequal groups in the study.

Here are some other minor questions

Page 6: We videotaped neither the record of daily routines nor the meetings with the groups of assistants - Then what did you videotape?

Page 7: Table 3 – It has two means but one SD…Thus, I guess SD should be indicated besides the Mean scores of Adult learning, Joint planning, Observation, Action/Practice/Reflection/Feedback in the Table 3

Page 7 & 8: The Call interview excerpts from the assistants should be presented in a more categorized and organized way.

Round 2

Reviewer 2 Report

The revised versions has been greatly improved. I think the authors have revised and corrected all the comments and answered to all the questions. as far as the authors could. Thus, I guess the revised versions now is ready for publication.